# The Impact of Exercise Intervention with Rhythmic Auditory Stimulation to Improve Gait and Mobility in Parkinson Disease: An Umbrella Review

**DOI:** 10.3390/brainsci11060685

**Published:** 2021-05-22

**Authors:** Roberta Forte, Nicoletta Tocci, Giuseppe De Vito

**Affiliations:** 1Department of Human Movement and Sports Sciences, University of Rome Foro Italico, 00135 Rome, Italy; 2Department of Biomedical Sciences, University of Padova, 35131 Padova, Italy; giuseppe.devito@unipd.it

**Keywords:** Parkinson, physical exercise, walking, mobility

## Abstract

Difficulties in walking, controlling balance, and performing activities of daily living are common problems encountered by individuals affected by Parkinson disease. Scientific evidence suggests that exercise performed with music or auditory or rhythmical cues facilitates movement and improves balance, gait, mobility, and activities of daily living (ADL) performance in patients with PD. The aim of this umbrella review was to summarize available high-quality evidence from systematic reviews and meta-analyses on the effectiveness of rhythmically cued exercise to improve gait, mobility, and ADL performance in individuals with PD. PubMed, Cochrane, and Embase databases were searched from January 2010 to October 2020 for systematic reviews and meta-analyses which had to be (1) written in English, (2) include studies on populations of males and females with PD of any age, (3) analyze outcomes related to gait, mobility, and ADL, and (4) apply exercise interventions with music or auditory or rhythmical cues. Two independent authors screened potentially eligible studies and assessed the methodological quality of the studies using the AMSTAR 2 tool. Four studies, two systematic reviews and meta-analyses, one a systematic review, and one a meta-analysis, were selected. Overall results indicated positive effects for gait and mobility of the use of rhythmic auditory cueing with exercise and suggested that it should be incorporated into a regular rehabilitation program for patients affected by PD. Nonetheless, more primary level research is needed to address the identified gaps regarding the application of this method to physical exercise interventions.

## 1. Introduction

Parkinson disease (PD) is the second most-common age-related neurodegenerative disease present in modern societies. According to recent research, the number of individuals affected by the disease has doubled between 1996 and 2016 to reach an estimated 6 million worldwide and will rise to 13 million people in 2040, suggesting a potential “pandemic PD affection” [1]. One of the main causes of such increment is the growth of the aging population. In fact, the disease is uncommon before 50 years of age, is prevalent at 60–70 years, and peaks between 85 and 89 years [2]. The causes of Parkinson disease are not completely known at present. Genetic factors may, in part, contribute to the development of the disease, but a series of environmental and occupational exposures linked to industrialization such as pesticides, solvents, or metals have also been identified as potential contributors [3]. The disease develops from a pathophysiologic loss or degeneration of dopaminergic neurons in the substantia nigra pars compacta and the development of neuronal Lewy bodies, inclusion bodies—abnormal aggregations of protein—that develop inside nerve cells affected by the condition. The diagnose of PD is based on the presence of symptoms gradually progressing and worsening over time. The most significant include tremor, rigidity, akinesia and bradykinesia, and postural instability, as well as freezing of gait, stooped posture, and reduced head and neck mobility. These are accompanied by significant psychological disorders of mood and cognition. All these manifestations significantly affect gait and general mobility in PD patients who typically show reduced gait speed, step length, increased gait asymmetry and double support time, and altered rhythmicity with increases in variability of step/stride time as well as step/stride length [4,5]. Gait problems clearly worsen with disease progression, resulting in a significant increase in the patient’s risk of falling and loss of the independence and quality of life [6].

Currently there is no effective cure for Parkinson disease and the treatments in use so far help alleviate mainly the motor symptoms of the disease but are not sufficient to restore motor function completely. These include pharmacological interventions, mainly through levodopa to increase the level of available dopamine, and surgical treatments with deep brain stimulation. Not all motor symptoms respond to pharmacological treatment. Tremor, bradykinesia, and akinesia improve following dopamine supplementation, and benefits are also observed in gait velocity and step and stride length. However, cadence, freezing of gait, balance, and axial rigidity are mainly unaffected and findings for other gait parameters remain controversial [7,8]. Similarly, non-motor symptoms such as cognition, attention, and mood are not influenced by the use of dopamine supplementation [9]. Moreover, the long-term pharmacological therapy, i.e., levodopa and levodopa sparing, can have severe effects on health and quality of life such as dyskinesis, loss of efficacy, and toxicity [10]. Similarly, as disease progresses, medications lose effectiveness [11], especially on gait parameters [12]. Deep brain stimulation, though providing some benefits for gait, is recommended for specific cases only.

Investigating alternative treatments to alleviate gait and mobility impairments is, therefore, of major importance in the interest of maintenance of patients’ health and quality of life. In this regard, physical exercise interventions are emerging as useful adjunct therapies to control both motor and non-motor symptoms in PD.

Exercise may benefit gait and mobility in patients with PD through the improvements in general health and physical and motor fitness or by triggering neuroprotective mechanisms and increase neuroplasticity. Though much remains to be uncovered, exercise may induce a protecting effect on the residual neurons of the nigrostriatal dopamine system or a restoring action of the dysfunctional cortico-basal ganglia motor control circuit. These benefits seem mediated by exercise-triggered production of endogenous neurotrophic factors [13,14,15]. It remains unclear, however, whether exercise can delay the progression of PD, but the short-term benefits of exercise training programs suggest this possibility [16].

As a result, besides traditional physiotherapy, a large variety of physical exercises, such as aerobic and strength exercise, treadmill training, aqua-fitness, dance, Taiichi, and Qigong, are being studied and tested for people with PD. Research findings support the practice of a large variety of physical exercise [17,18,19,20] to manage motor and possibly non-motor symptoms and to improve physical fitness and functional mobility and reduce the risk of falling in individuals with PD.

Among the multitude of interventions, the use of music and rhythmically cued exercise or rhythmic auditory stimulation (RAS) is gaining wide attention for its positive effects not only on mobility and gait, but on general motor coordination in PD patients (for review, see [21]).

Recent research shows the benefits of using external sensory cues demonstrating a significant decrease in the use of levodopa equivalent to 1-year follow-up in a group of patients following movement therapy with external sensory cues [22]. Similarly, other data indicate significant improvements in ADL and motor performance following treatment with external sensory cues during an “off” medication period [23,24]. This method of exercise seems to facilitate the organization and the initiation of movement due to the extensive connections between auditory and motor areas in the brain [25] and appears to benefit particularly gait disorders, freezing [26,27,28], and other gait- and balance-related difficulties [29]. Although the underlying mechanisms of functioning of rhythmic cueing are not completely known, a possible explanation is the close proximity of the areas of the brain implicated in the perception of rhythm with those controlling movement [30]. Auditory cues appear particularly effective with respect to other sensory stimuli, such as visual or tactile, as faster and more precise [25,31]. Sound can increase motor neurons’ excitability involving auditory-motor circuitry at the level of the reticulo-spinal tract [32], helping anticipatory motor control patterns at the brain stem and spinal cord, reducing the time for a muscle to activate a motor command. Brain imaging studies show that music and rhythm perception produce changes in the auditory-motor network, which facilitates movements and can be taken advantage of in physical therapy for PD [33]. External rhythm can help the individual affected by PD to synchronize ground contact and lift-off times when walking [34]. The cyclic characteristic of rhythmic auditory cueing appears to effectively control variability in musculoskeletal activation patterns, resulting in more efficient motor unit recruitment, facilitating joint movement and grading movement time [33].

The auditory cue can take many forms, such as music, the beat of a metronome, vocalization such as counting, or any other type of rhythmic stimulation. People usually synchronize their actions through an innate rhythmic entrainment [30]. This process is postulated to depend on the complex interaction of several structures of the subcortico-thalamo-cortical network including the cerebellum, basal ganglia, pre-supplementary motor area, and supplementary motor area [32]. All these areas are negatively affected during the progression of PD, causing, among other impairments, the loss of the ability to produce a normal steady gait rhythm. RAS aims to the synchronization of movement (i.e., walking) to an external sound, the tempo of which is increased or decreased in order to find the optimal pace [35]. RAS seems to create fast and predictable temporal synchronization patterns between sensory input and motor output, which makes possible the anticipation of movement, for example of the next steps when walking, helping to make internal rhythm steady and possibly replacing the dysfunction in internal timing regulation. The cyclic characteristic of rhythmic auditory cueing appears to effectively control variability in musculoskeletal activation patterns, resulting in more efficient motor unit recruitment, facilitating joint movement and grading movement time [33]. For the purpose of the present work, only RAS in its different forms was considered, excluding the use of music therapy and dance, which represent very specific forms of intervention recognized to generate emotional and psychological reactions not generally provoked by a simple acoustic sound [27].

The effectiveness of RAS has been summarized in several systematic reviews and meta-analyses. However, the literature so far appears confusing as reviews mix interventions with different cues (visual, proprioceptive), physical therapies (exercise with or without auditory cue), or various neurological conditions [36,37,38,39,40,41]. Moreover, guidelines on the most appropriate use of RAS are still incomplete. Therefore, the aim of the present review was to synthesize, through an umbrella review, the current evidence presented in systematic reviews and meta-analyses on the effects of exercise interventions’ RAS only for gait and mobility disorders in patients with Parkinson disease. The present review represents a step forward in the identification of specific evidence to guide future development of guidelines for RAS interventions to reduce mobility and gait disorders of Parkinson patients.

## 2. Materials and Methods

### 2.1. Study Design

The present umbrella review was conducted according to the PRISMA statement, which involved critically examining the literature, starting from a synthesis of the previously published, second-level research, and performing a critical review of all available evidence.

### 2.2. Search Strategy and Study Selection

PubMed, Cochrane, and Embase electronic databases were comprehensively searched for systematic reviews and meta-analyses published from January 2010 to October 2020. To be included in the present review, these had to meet the following criteria: (1) deal with individuals with Parkinson disease of any age and level of severity; (2) describe effects of interventions of physical exercise/activity or physical therapy applying music or any other rhythmical or auditory stimulus; (3) assess effects on gait and mobility including freezing of gait and ADL; (4) reviews published in English; and (5) reviews including only randomized controlled trials, controlled clinical trials, and quasi-randomized controlled trials. Music therapy and dance in any form were excluded as considered specific methods of activity not comparable with physical exercise or physical activity. Sensory stimuli that did not concern the objective of the review were also excluded. The inclusion and exclusion criteria are shown in Table 1.

The following string, adapted for use in the different databases, was used by the authors to conduct the search: ((Parkinson disease) AND (music* OR rhythmic movement OR music-based intervention OR acoustic stimulation OR auditory stimulation OR cueing OR cue) AND (gait OR mobility OR balance OR postural instability OR fall* OR risk of fall*) NOT (dementia OR cognitive impairment OR frail)).

### 2.3. Quality Assessment

The methodological quality of each eligible study was assessed independently by two investigators using the AMSTAR 2 questionnaire [42], which evaluates the methodological quality against 16 questions. The most critical domains determining the quality of the review concern protocol registration before the commencement of the review (item 2); adequacy of literature search (item 4); justification for excluding individual studies (item 7); risk of bias from individual studies being included in the review (item 9); appropriateness of meta-analytical methods (item 11); consideration of the risk of bias when interpreting the results of the review (item 13); and assessment of the presence and likely impact of publication bias (item 15). Four levels of classification were obtained: ‘high-quality’ for none or one non-critical weakness; ‘moderate-quality’ for presence of more than one non-critical weakness; ‘low-quality’ for presence of one critical flaw with or without non-critical weaknesses; ‘critically low-quality’ for presence of more than one critical flaw with or without non-critical weaknesses. Any disagreement in rating between the investigators was resolved by discussion until consensus was reached.

### 2.4. Data Extraction

Data extraction was performed independently by two researchers according to standard guidelines (i.e., JBI, Cochrane). Information gathered concerned: (1) the study details, authors names, year of publication, the objective of the study; (2) participants’ characteristics, number, age, gender, disease level; (3) the description of the intervention, type of exercise and auditory stimulus, duration, frequency; (4) the number of included studies and type of included studies; (5) the appraisal instruments used and results; (6) the analysis performed; (7) the outcomes measured; (8) the results of the study; and (9) the conclusions. Any incongruity in the data extracted by the investigators was resolved by discussion between reviewers until consensus was reached.

## 3. Results

### 3.1. Study Selection

There were 395 papers retrieved (173 from PUBMed, 209 from Embase, 13 from Cochrane). After 181 duplicates were removed, title and abstract screening was performed on 214 studies, which led to 170 studies being judged as irrelevant. Reference lists of selected papers were checked for relevant literature and no further study was retrieved. Therefore, 44 studies underwent full-text screening, which resulted in a further 40 studies excluded. Consequently, four studies were found eligible to proceed to data extraction. Any disagreement was resolved by discussion between reviewers until a consensus was reached (Figure 1, PRISMA flow chart).

### 3.2. Study Characteristics

The eligible studies were also screened for overlapping references according to Pieper et al. [43]. The authors suggest performing this procedure for umbrella reviews when the selected works present the same or similar topics and interventions to avoid the inclusion of primary studies more than once and the risk of biased findings. Therefore, the following formula was used for calculating the corrected covered area, CCA (%) = *N*-r/rc-r, where N = number of included publications (sum of checked boxes), r = number of rows (primary publications), and c = number of columns (number of reviews). Three levels of overlapping may by identified: slight 0–5, moderate 6–10, high 11–15, and very high >15. A slight level of overlapping was observed for the present analysis.

With respect to the methodological quality of the included reviews, one study resulted as moderate quality [44], two as high quality [45,46], and one as low quality [26]. The items where the studies did not meet the quality requirements were: registering the protocol before the commencement of the review (item 2, *n* = 4); explaining the selection of the study designs for inclusion (item 3, *n* = 1); consulting content experts in the field (item 4, *n* = 4); providing a list of excluded studies and justifying the exclusions (item 7, *n* = 1 no, 1 partial yes); assessing the risk of bias of individual studies included in the review (item 9, *n* = 1); reporting the sources of funding of individual studies (item 10, *n* = 4); and investigating and discussing publication bias (item 15, *n* = 2). All reviews considered heterogeneity as a source of variation between the different studies. Regarding the consideration of the risk of bias when interpreting the results of the review (item 13), though not all authors directly declared it, the issue was limited by including studies of levels II to IV of the National Health and Medical Research Council hierarchy of evidence [46,47] or following careful evaluation through a standardized table [26]. Complete agreement was reached between researchers on the quality assigned to the studies.

A narrative synthesis of the data extracted from the eligible studies is presented in Table 2, Table 3 and Table 4. Briefly, studies were published between 2013 and 2018, in Canada [26], Australia [44], Germany [45], and Brazil [46]. Two studies were systematic reviews with meta-analyses [43,44], one a systematic review [46], and one a meta-analysis [26]. In the included reviews, one study comprised RCTs only [43], two included RCTs, CCTs, and QRCT [45,46], and one did not limit the study design of the included studies [26]. A total of 2338 patients were gathered, 1410 male and 1506 female, apart from the studies included in all review not specifying or distinguishing the gender of the involved participants, with an age range 24 to 83 years (younger participants were healthy controls; [45]) with a disease stage II–IV [44,46] or unclearly reported [26], and duration of the disease of 9 months [46] to 14 years [45]. One study mentioned that the assessment and treatment were performed during the “on” period of medication [46].

All the reviews [26,44,45,46] presented the outcomes of studies involving different types of sensory stimuli and/or their combination. This umbrella review reports the results related to auditory stimuli only. Regarding the interventions, the auditory stimuli varied, including metronome, words, clunk or ding sounds [44]; metronome, musical beats, percussions [45]; and metronome, prototype rhythmic auditory device, pace, a click tone simulator, music on a synthesizer, RAS music, and RAS [26]. In the study by Rocha et al. [46], the nature of the auditory stimuli applied in the included studies was not specified, except for the mentioning of “words”. In two studies, the tempi of presentation of the auditory stimuli were reported to range overall from −5% to +25% of preferred cadence [26,44,45], or 30 to 150 metronome bpm [26,45]. Exercises included predominantly walking overground and walking on treadmill but also stepping or performance of ADLs (e.g., sit to stand, turning, and so on) with concomitant auditory stimulation [46]. It could also be deduced that some studies investigated acute effects of RAS (i.e., RAS with or without training; [45].

Regarding exercise volume, information was often incomplete. Ghai et al. [45] reported session durations ranging from 20 min to 1 h, with three to five sessions per week, without reporting the overall number of weeks. Cassimatis et al. [44] reported sessions’ durations ranging from 45 to 90 min, for two to three days per week, for an overall duration of 2 to 6 weeks. Rocha et al. [46], reported two to four sessions per week, for 3 to 8 weeks, for 20 min to 1 h. Spaulding et al. [26] reported three to seven sessions per week, without reporting the overall number of weeks, and 4 to 8 weeks, without reporting the number of sessions per weeks. Rocha et al. [46] reported two to seven sessions for 2–4 weeks, with each session lasting approximately from 20 min to 1 h.

### 3.3. Study Outcomes

The measured outcomes are summarized in Table 3. Briefly, these included the assessment of ADLs through the UPDRS II scale [44], which, in part II and III, specifically assesses performance of ADL such as walking, turning, writing, and rising from sitting. In the other studies, the following gait parameters were assessed: gait speed, stride length, cadence [26,45,46], double limb support time and turn time [45], and step length [46].

Summary Estimates and Related 95% CI *p*-value. RAS = Rhythmic Auditory Stimulation; ES = Effect size; CI = confidence interval; g = Hedge’s g adjusted effect sizes; I^2^ test of heterogeneity.

Overall studies reported significant positive effects of the application of auditory stimuli during training on the measured parameters. Details are reported in Table 3 and Table 4 Briefly, for the analyzed spatiotemporal gait parameters, of the three studies measuring such outcomes, all reported gait speed to positively improve (medium effect g: 0.5, [26]; *p* < 0.001, [46]; small effect g: 0.23, *p* < 0.05, [45]). One study reported increase in gait cadence with medium effect, g: 0.56 [26]; two in stride length with medium effect, (g: 0.5, [26]), and small effect (g: 0.4, *p* < 0.01, [45]); and one in step length (*p* < 0.05, [46]). Moreover, a small negative effect (g: −0.05, *p* < 0.01) on gait cadence with a reduction in the number of steps was reported by Ghai et al. [45], and Rocha et al. [46] reported no benefits for stride length and cadence with auditory stimulation. It should be mentioned that the authors did not clarify which of the effects on cadence should be considered as improvements, positive effects (increments), or negative effects (decrements).

Among the reported studies, one study only [46] evaluated changes in the frequency of FOG. Even if this study used combined cues (visual and auditory), a significant reduction in freezing episodes after the intervention was observed (*p* = 0.04).

Finally, one study [44] showed that external sensory-cued therapy improves ADL performance for individuals with PD both after treatment and at follow-up. The treatment with auditory cues showed a statistically significantly improvement on average for the UPDRS II after treatment (*p* = 0.011) and at follow-up (*p* = 0.001) under the random-effect model [MD = −2.27 (95% CI = −4.03 to −0.52); I^2^ = 0%, *p* = 0.512; and MD = −3.57 (95% CI = −5.72 to −1.42); I^2^ = 0%, *p* = 0.531, respectively).

## 4. Discussion

This review summarizes the findings of systematic reviews and meta-analyses on the benefits of using RAS during exercise to improve gait and mobility in patients with Parkinson disease. Despite the small number of available reviews and meta-analyses specific to RAS and physical exercise, the results confirmed the general view that RAS presented in different forms with exercise improves gait parameters such as gait speed and stride length and facilitates overall mobility.

The changes in gait parameters following the RAS intervention were consistent across the selected studies (step length (z = 2.15, *p* < 0.05 [46]), stride length (g: 42 [45]; g: 0.50 [26]; z = 6.73, *p* < 0.01 [46]), gait velocity (0.54 [26]; g: 23 [45]; z = 5.41, *p* < 0.001 [48]), and cadence (g: 0.56, [26]; g: −0.05, [45])), for which non-significant effect was found by Rocha et al. [46]. The analysis of the “on” and “off” state of medication seemed to confirm that the effects of RAS do not seem to depend on dopaminergic medication, as previously reported [21]. In fact, Ghai et al. [45] found for stride length in both “on” and “off” state a large-effect size (g: 0.86 and g: 0.96, respectively), while for gait speed, there was positive, small-effect size for “on” (g: 0.43) and medium for “off” (g: 0.55).

In this respect, some comments are needed on the relationship between gait speed, stride length, and cadence, which are all critically affected by PD [5], and the effects of exercise. It is known that spatiotemporal parameters of gait give objective information on the global performance of this vital action of human life, and some more than others are general predictors of health [47]. While improvements following exercise on gait speed, step length, and stride length can all be interpreted as positive, the same cannot be said for cadence. Gait speed results from the combination of cadence and stride length. Cadence, or the number of steps per minute, is generally increased in aging to compensate for the reduction in stride length to maintain gait velocity, and increases in cadence may negatively affect stability while walking [48]. In PD, depending on the stage of the disease, cadence can either decrease or increase to become the typical “shuffled steps”, with short stride length and very frequent steps [49].

Regarding the UPDRS II scale, though it is not a direct evaluation of walking and mobility, but an evaluation of the effects of motor and non-motor symptoms of PD on various motor experiences of daily living including walking and mobility, it gives necessary information on the level of the disease, which may be overlooked by an objective spatiotemporal gait assessment. Indeed, UPDRS II scores demonstrated significant improvement following interventions with RAS (z = 2.5, *p* < 0.05 [44]).

Two main hypotheses have been formulated to explain how RAS may compensate the timing deficit in PD, the compensation and restoration hypotheses (for review, see [50]). The compensation theory proposes that areas other than the basal ganglia are activated to substitute the basal ganglia in rhythm perception and are linked to the internal processing of timing. Movement with acoustic stimulation has been found to increase activation in the lateral premotor cortex and the cerebellum, seemingly recruited to compensate for the impaired basal-ganglia-cortical circuitry, a sort of bypass of the striato-thalamocortical circuitry (i.e., use of internal timing). The restoration hypothesis suggests that instead of bypassing damaged circuitry, RAS may improve the basal ganglia neuronal activity through enhancement of neural oscillations, in particular, of beta frequency [51]. These are normally observed in deep brain recordings in humans and play a role in the maintenance of a specific state [52]. However, in individuals with PD an excessive beta waves’ oscillatory synchrony in the cortico-basal ganglia circuits has been found, which supposedly disturbs neuronal communication between the motor cortex and basal ganglia and produces the symptoms of akinesia and rigidity. Both hypotheses are supported by findings. However, research is still needed to indicate if one of the two hypotheses or both play a determinant role in the effectiveness of auditory stimulation for motor improvements in PD.

Regarding the necessary evidence to build recommendations for the use of RAS for gait and mobility exercise in PD patients, the reviewed studies did not provide a complete range of information. It is well known that for exercise to be effective the patient must adhere to the training principle of overload (duration of exercise sessions and of whole intervention, frequency, and intensity), specificity (of the exercise in relation to the objectives), and individual differences.

The existing guidelines of exercise for PD patients are very generic. For example, the American College of Sports Medicine recommends 20 to 60 min of aerobic exercise, such as walking, cycling, and dancing, 3 to 5 days per week, and strength and stretching exercises for 2–3 days per week. Other forms of recommended exercise include exercise specifically targeting balance and gait [53]. However, no specific prescription exists for the application of rhythmic auditory stimulation.

Indeed, none of the included reviews was able to derive sufficient information from their primary studies to draw up conclusions related to such principles. One study only [45] gathered some prescriptions in terms of tempo of the stimuli, duration, and weekly frequency of the exercise session. Based on their extensive sub-analyses of the reviewed studies, the authors suggest the use of tempi of auditory stimulation ranging from ±10% of the preferred cadence to obtain optimal results for gait variability. Regarding duration, sessions should be 25–40 min, as longer periods may induce fatigue and produce deleterious effects. Optimal weekly frequency seems to range between 3 to 5 days per week. However, the authors themselves warn about the limit of such findings, due to the large heterogeneity of the data.

Strictly regarding the use of RAS for gait and mobility, in spite of the apparent large number of reviews and meta-analyses, very little material was available. This resulted in a lack of information on to how to modulate the “overload” of RAS. An additional challenge was posed by the identification and definition of the reviewed outcomes. While kinematic measures of gait are clearly defined and standardized measures, difficulties arise when meta-analyses or systematic reviews have similar but not identical definitions of outcomes such as mobility, which may include a wide variety of functional tasks and related measures.

The present review also highlights limitations in the present literature on the use of RAS for gait-related impairment and may help in planning future research on this extremely important issue. The first-level studies summarized in the reviews and meta-analyses reviewed in the present work appeared extremely heterogeneous in all their components, participants, levels of the disease, state of medication, and proposed exercise interventions, with some studies evaluating the acute effects of exercise and RAS and others evaluating the long-term with training, and did not allow establishing the optimal elements of exercise prescription.

Intervention studies are increasing in number. However, there are still too many variations in designs. This is an area that needs addressing in future research: the design of specific exercise interventions aiming at determining the guidelines for optimal exercise application for PD patients. None of the interventions seemed to design the exercise following the key training principles or adapting them to the specificity of the disease. Some standardization has been proposed in terms of tempo of stimuli with respect to preferred cadence. Though the method is highly promising, it is extremely difficult to reach a definite conclusion to establish guidelines for the optimal use of RAS in PD patients. Future research should aim to fill these gaps with more methodologically sound intervention studies.

It also needs to be mentioned that despite the selected studies being carefully evaluated for methodological quality, these often presented inadequate description of methods or a confusing presentation of results, making difficult a full assessment and a detailed synthesis. Most studies did not explicitly state several of the requested issues of the quality assessment without necessarily being of low quality and, on the other hand, high-quality studies, responding to all the quality assessment issues, presented flaws in the manuscript presentation. Some studies did not describe information of paramount importance regarding the applied exercise.

Some negative effects of RAS on gait stability parameters have also been described, suggesting that RAS should be “calibrated” to the subjects’ conditions and used appropriately. In fact, when the auditory stimulus was slower (−20%) or faster than a preferred cadence, if no instruction was given to coordinate gait with the external beat or if combining auditory with tactile stimuli, detrimental effects on gait parameters were described, possibly due to an excessive cognitive load exceeding the individual’s attention capacity [54]. In addition, the use of the auditory cue should not make us forget the fundamental guidelines of exercise and motor skill training. It is well known that exercise improves gait and balance through a combination of central mechanisms of control and indirectly through fitness-related adaptations. Therefore, besides correctly applying the auditory cue, to optimize the effects of the proposed training for walking and balance control, sufficient overload (i.e., n. of repetitions, frequency, duration) of task-specific exercise should be performed to induce significant modifications. Moreover, to have a better “transfer” effect and to prepare the individual to face the complex and manifold daily life situations, possibly a multi-component approach to exercise should be also considered. Exercise should ideally be designed following the principle of variability, stimulating physical fitness in all its components as well as in cognitive, emotional functioning, functional abilities, and a long-lasting interest in physical activity.

## 5. Conclusions

Recent systematic reviews and meta-analyses summarizing results of several intervention studies led to the general conclusion that RAS is a feasible and valid tool to significantly improve balance and gait parameters such as velocity, cadence, and stride length [26,45,46,55]. The use of an auditory external stimulus enhances the capacity to maintain a steady rhythm while walking and induce more stable walking through the reduction in the asymmetry of muscle activation. In addition, improvements in the UPDRS scale and in freezing have been also observed but with differences depending on the level of the disease.

This method certainly merits further investigation and future research to reinforce the findings with specific design to identify optimal dosage in term of frequency, intensity, duration, and optimal type of exercise. Developing effective non-medical intervention to relieve gait and balance impairments would represent a great advancement in Parkinson disease therapy.

## 6. Limitations

Limitations of the present work include the narrative nature of the review and the very limited number of included systematic reviews and meta-analyses, which did not allow us to clarify the evidence necessary to establish guidelines for practice. Though three main databases were searched, and the search string was as comprehensive as possible, there is the possibility that some research may have not been included.

## Figures and Tables

**Figure 1 brainsci-11-00685-f001:**
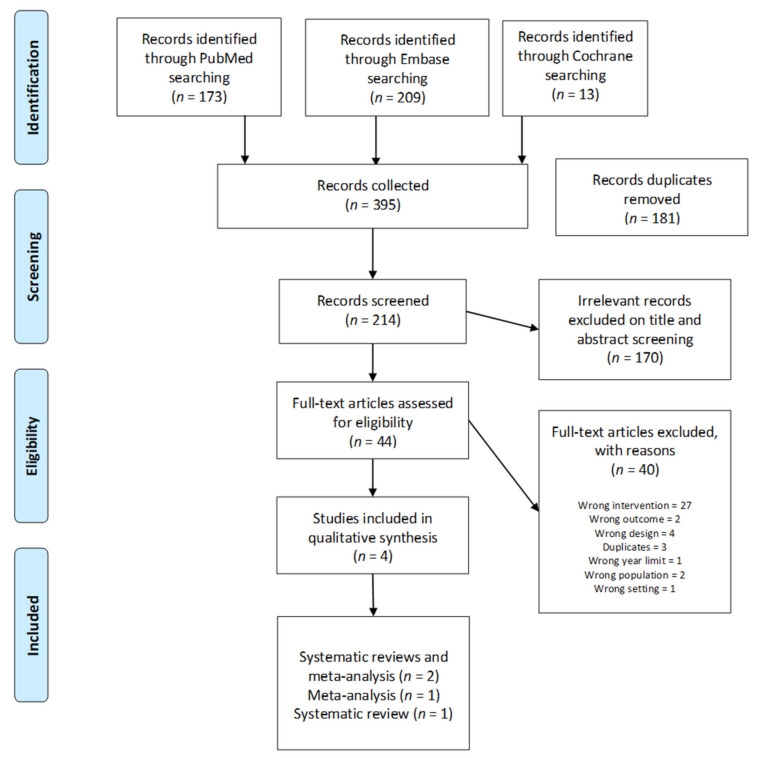
PRISMA flow chart of selected studies.

**Table 1 brainsci-11-00685-t001:** Inclusion and exclusion criteria for reviews and meta-analyses of studies.

Inclusion Criteria for Reviews and Meta-Analyses	Exclusion Criteria for Reviews and Meta-Analyses
Studies on individuals with Parkinson disease of any age and level of severity	Studies with frail participants or with other neurological conditions, or cognitive impairment
Studies on physical exercise/activity interventions, applying music or other rhythmical or auditory stimulus	Studies on any form of dance or music therapy
Studies with primary outcomes related to mobility and gait (including FOG) and activities of daily living (ADL)	Studies with primary aims other than gait, mobility, and ADL
Studies in English	Narrative reviews
Studies published between 2010 and 2020	Studies on other type of stimulus/their combination with auditory stimulus
Reviews including RCTs, CCTs and QRCT	

FOG = Freezing of Gait; RCT = randomized controlled trials; CCT = controlled clinical trials; QRCT = quasi randomized controlled trials.

**Table 2 brainsci-11-00685-t002:** Characteristics of included studies and overall results.

Authors, Year	Aim of Study, Type of Study	Number of Included Studies, Type of Included Study	Participants: Number, Gender, Age Range, Disease Level/ Duration	Outcome	Intervention Protocols, Duration (Overall and Weekly Frequency)
Cassimatis et al., 2016 [44]	To verify the effects of external sensory cues in improving functional performance of ADL, walking, and daily tasks in PD;	3 studies were included in the meta-analysis of pre-post treatment	112, M&F, 63–73 years, II–IV	UPDRS II	45–90 min 2–3 days per week 2–6 weeks Applied auditory stimuli: metronome, words, clunk, ding sounds Proposed Exercises with RAS: overground walking, treadmill walking, turning, stepping, sit to stand, performance of ADLs
	Systematic review and meta-analysis	RCTs level II of the NHMRC Hierarchy of Evidence (National Health and Medical Research Council)			
Ghai et al., 2018 [45]	To analyze the effects of rhythmic auditory cueing on gait and postural performance in patients affected by Parkinson disease.	51, RCT, CRCT, CCT	1892, M = 1089, F = 745; 24–79 years; Duration of disease 3–14 years	Gait and postural parameters: Gait velocity Stride length Cadence Double limb support time Turn time	RAS and cadence: Preferred; +5%, +10%, +15%, +20% +25%; −10%; 105–144 bpm; 96–100 bpm. Metronome: preferred cadence, 80–124 steps min; −10, −20, +10, +20, +25% of preferred cadence;metronome embedded in music;30, 60, 90, 120, 150 bpm; Music: beat rhythm adjusted −10, +5, +10% preferred cadence;Musical instruments: Piano tones; percussions; Other sounds: Clicking sound generated with gait step; synthesized gravel step sound corresponding to plantar force; beep at preferred cadence; high pitched beep at preferred cadence;
	Systematic review and meta-analysis	7 Randomized controlled trials (RCT), 44 controlled clinical trials (CCT)			
Rocha et al., 2014 [46]	Evaluate the benefits of external cues on the gait of PD patients and their impact on the quality of life, freezing and psychomotor performance	7 studies: 6 RCTs, 1 QRCT	204, M&F mean age 68.8 years. The Hoehn Yahr, mean stage was 2.4.	Gait (step length, stride length, speed, cadence), freezing, psychomotor (UPDRS)	The authors did not include stimulus intensity and type of auditory cues in the session.
	Systematic review of randomized clinical trials (RCTs) and quasi-randomized clinical trials (QRCTs)				Treadmill + visual cues × overground + visual cues × control (no intervention) 18 therapies 6 weeks 3 × week 30 min
					Conventional therapy + overground + visual cues × conventional therapy 20 therapies 01 h
					Treadmill + auditory cues × treadmill × home gait training 12 therapies 4 weeks 3 × week 30 min
					Overground + auditory cues × overground gait training × control (no intervention) NA 3 weeks NA 30 min Overground +verbal instruction × control (no intervention) 10 therapies 2 weeks 5 × week NA
Spaulding et al., 2013 [26]	To compare the relative efficacy of visual versus auditory cueing on gait among individuals with Parkinson disease (PD).	25 peer-reviewed publications.	718, M&F, age mean 61.7–73.3. Duration of PD (mean) 6.5–10.0. MMSE > 24–29.0 (range 27–30) HY Stage (mean) 1 or 2–18/4. Medication On/Off. UPDRS range 10.9-44	Kinematic gait parameters (cadence, velocity and stride length)	Prototype rhythmic auditory device, time of intervention to posttest: Immediate Pacer: Immediate.
	Studies that reported pre-and post-outcome measures of gait parameters.				

RCT = randomized controlled trials; CRCT = cluster randomized controlled trials; CCT = controlled clinical trials; RAS = Rhythmical Auditory Stimulation; g = Hedge’s g significance; Hedges’ g; ES = Effect Size; CI = Confidence Interval.

**Table 3 brainsci-11-00685-t003:** Methods used in the included studies.

Authors, Year	Appraisal of Methodological Quality, Score	Statistical Method	Results/Summary Statistics (95% CI)
Cassimatis et al., 2016 [44]	PEDRO scale 4–8 low to moderate RCT	Random effect model (R the metafor package)	UPDRS II: post-treatment RE model: total (95% Cl) MD (95%CI) −2.27 (−4.03 to −052) Heterogeneity: Ƭ^2^ = 0; 2 = 1.3380, d.f. = 2 (*p* = 0.5122); I^2^ = 0% Test for overall effect: Z = −2.5353 (*p* = 0.0112)
	high (7 or higher), moderate (4, 5 or 6) and low (less than 4) quality	Mean scores, associated SDs and sample sizes for treatment and control groups were extracted and analyzed. Mean difference (MD) in outcome between groups after treatment was computed. The MD could be applied as all studies used the same outcome measure. Results were considered statistically significant if the 95% confidence intervals (CIs) of the forest plot did not include 0. The statistical significance is summarized as a *p*-value, *p* < 0.05 considered statistically significant.	
	Systematic review of RCTs of level II (low risk of bias) of the NHMRC Hierarchy of Evidence (National Health and Medical Research Council)	Heterogeneity was tested with I^2^-test, values greater than 40% interpreted as considerable heterogeneity. A *p*-value < 0.05 indicates significant heterogeneity of intervention effects	
Ghai et al., 2018 [45]	PEDro quality scale Included only ≥4, mean 5.4 (fair) range 4–8	Random and fixed effect meta-analysis	Sub-group analyses were performed in case of unexplained heterogeneity (see text) Gait velocity (34 studies): small ES (g: 0.23, 95% C.I: 0.1 to 0.3) with substantial heterogeneity (I^2^: 87.4%, *p* > 0.01). On-off medication: “off”: positive small ES, (g: 0.43, 95% C.I: 0.11 to 0.75, I^2^: 18.8%, *p* = 0.29), negligible heterogeneity; “on”: positive medium ES (g: 0.55, 95% C.I: 0.23 to 0.87, I^2^: 0.0%, *p* = 0.44), negligible heterogeneity. Use of treadmill: positive large ES, negligible heterogeneity (g: 1.0, 95% C.I: 0.33 to 1.67, I^2^: 24.6%, *p* = 0.24). “Fast” and “Slow” auditory cueing: positive medium ES for “fast”, negligible heterogeneity (g: 0.7, 95% C.I: 0.50 to 0.89, I^2^: 0.0%, *p* = 0.44), negative small ES for “slow” group, negligible heterogeneity (g: −0.24, 95% C.I: 10.51 to 0.19, I^2^: 23.53%, *p* = 0.24). Acute effect of auditory stimulus (10 studies): negative small ES (g: −0.34, 95% C.I: −0.5 to −0.18, I2: 85.9%, *p* < 0.01), substantial heterogeneity. Disease duration: <9 years positive small ES (g: 0.16, 95% C.I: −0.12 to 0.44, I^2^: 0%, *p* = 0.56), negligible heterogeneity; >9 years negative small ES (g: −0.37, 95% C.I: −0.62 to −0.13, I^2^: 91%, *p* < 0.01), substantial heterogeneity. Training effect of RAS: positive medium ES with moderate heterogeneity (g: 0.64, 95% C.I: 0.37 to 0.92, I^2^: 36.08%, *p* = 0.34). Session duration: 30–45 min, positive medium ES with moderate heterogeneity (g: 0.52, 95% C.I: 0.38 to 0.66, I^2^: 33.8%, *p* > 0.05). 20 min, positive large ES with substantial heterogeneity (g: 1.09, 95% C.I: 0.7 to 1.47, I^2^: 80.9%, *p* < 0.01). Training duration: <5 weeks positive medium ES, substantial heterogeneity (g: 0.73, 95% C.I: 0.31 to 1.14, I^2^: 21.3%, *p* > 0.05). >5 weeks positive small ES, negligible heterogeneity (g: 0.46, 95% C.I: 0.2 to 0.72, I^2^: 0%, *p* > 0.05) Stride length (35 studies): positive small ES (g: 0.42, 95% C.I: 0.35 to 0.5, I^2^: 85.05%, *p* < 0.01) substantial heterogeneity; On-off medication: large positive ES for both “on” and “off” (g: 0.86, 95% C.I: 0.52 to 1.2, I^2^: 0%, *p* = 0.64), (g: 0.96, 95% C.I: 0.59 to 1.34, I^2^: 0%, *p* = 0.39), negligible heterogeneity. “Fast” and “Slow” auditory cueing: “fast” positive small ES (g: 0.30, 95% C.I: 0.08 to 0.52, I^2^: 0%, *p* = 0.7), negligible heterogeneity; “slow” positive medium ES (g: 0.69, 95% C.I: 0.35 to 1.03, I^2^: 20.03%, *p* = 0.29), negligible heterogeneity. Un-modulated RAS (23 studies): positive small ES (g: 0.35, 95% C.I: 0.22 to 0.48, I^2^: 35.3%, *p* = 0.04) moderate heterogeneity: Acute effect: a positive small ES (g: 0.12, 95% C.I: −0.11 to 0.35, I^2^: 0%, *p* = 0.72) negligible heterogeneity; Training effect: small EF (g: 0.37, 95% C.I: 0.23 to 0.51, I^2^: 26.2%, *p* = 0.16), moderate heterogeneity 30 min positive small ES (g: 0.36, 95% C.I: 0.21 to 0.51, I^2^: 42.5%, *p* = 0.07), moderate heterogeneity. Training frequency: <5 weekly session, small positive ES (g: 0.39, 95% C.I: 0.08 to 0.7, I^2^: 0%, *p* < 0.11), negligible heterogeneity. >5 weekly sessions, small positive ES (g: 0.4, 95% C.I: 0.1 to 0.68, I^2^: 0%, *p* > 0.5), negligible heterogeneity. Cadence (30 studies): negative small EF (g: −0.05, 95% C.I: −0.13 to 0.03, I^2^: 93.6%, *p* < 0.01) with substantial heterogeneity “On” and “off” medication: “off” negative small ES (g: −0.10, 95% C.I: −0.46 to −0.25, I^2^: 81.97%, *p* = 0.01), substantial heterogeneity; “on” positive small ES (g: −0.13, 95% C.I: −0.20 to 0.46, I^2^: 89.69%, *p* < 0.01), substantial heterogeneity. Fast-slow tempo: “fast” stimuli in less severe patients showed positive medium EF (g: 0.61, 95% C.I: 0.25 to 0.94, I^2^: 0%, *p* = 0.49), negligible heterogeneity; in severe patients, positive large ES (g: 1.4, 95% C.I: 0.89 to 1.91, I^2^: 27.6%, *p* = 0.24), moderate heterogeneity. “slow” <9 years duration (less severe) showed negative medium ES (g: −0.5, 95% C.I: −0.97 to −0.04, I^2^: 0%, *p* = 0.93), negligible heterogeneity; >9 years (severe) negative large ES (g: −1.54, 95% C.I: −2.06 to −1.02, I^2^: 0%, *p* = 0.37), substantial heterogeneity. Preferred cadence: positive small ES (g: 0.17, 95% C.I: 0.01 to 0.32, I^2^: 90.5%, *p* < 0.01), substantial heterogeneity. Acute effect: positive small EF (g: 0.30, 95% C.I: 0.07 to 0.53, I^2^: 0%, *p* = 0.45), negligible heterogeneity; Training effect: small negative EF (g: 0.04, 95% C.I: −0.1 to 0.2, I^2^: 93.6%, *p* < 0.01), substantial heterogeneity; Training of 30 min: small EF (g: 0.09, 95% C.I: −0.06 to 0.25, I2: 95.5%, *p* < 0.01), substantial heterogeneity. Weekly frequency: <5 weekly, positive medium ES (g: 0.65, 95% C.I: 0.33 to 0.96, I^2^: 0%, *p* = 0.94), negligible heterogeneity. >5 days weekly showed negative small ES (g: −0.22, 95% C.I: −1.16 to 0.71, I^2^: 23.6%, *p* > 0.05), negligible heterogeneity. Double limb support time (8 studies): positive medium ES (g: 0.5, 95% C.I: 0.34 to 0.67, I2: 93.46%, *p* < 0.01), substantial heterogeneity. “fast”-“slow” tempo”: fast shows a positive small ES (g: 0.46, 95% C.I: 0.05 to 0.87, I^2^: 92.3%, *p* < 0.01), negligible heterogeneity. Slow shows small positive ES (g: 0.33, 95% C.I: −0.18 to 0.85, I^2^: 92.8%, *p* < 0.01), substantial heterogeneity. Preferred cadence: reduction in double limb support phase, medium negative ES (g: −0.56, 95% C.I: −0.9 to −0.22, I^2^: 0%, *p* = 0.72), negligible heterogeneity. Turn time (3 studies): In freezers negative large ES (g: −2.08, 95% C.I: −2.5 to −1.66, I^2^: 93.7%, *p* < 0.01), substantial heterogeneity. In non-freezers negative large ES (g: −2.3, 95% C.I: −2.71 to −1.88, I^2^: 87.67%, *p* < 0.01), substantial heterogeneity.
		Meta-analysis performed on pooled homogenous studies (CMA V2.0, USA).	
		Heterogeneity was assessed using I^2^ statistics with 0%, 25%, 75% interpreted as negligible, moderate and substantial heterogeneity, respectively	
		The ES were adjusted and reported as Hedge’s g interpreted as 0 = no change, 0.2 = small effect, 0.5 = medium effect, 0.8 = large effect, negative effect size = negative change.	
		Meta-analysis reports indicating heterogeneity among studies were evaluated to determine the reason of heterogeneity, and the included studies were then pooled separately and analyzed again	
		Publication bias was analyzed by plotting a Hedge’s g against standard error *p* < 0.05 was adopted.	
Rocha et al., 2014 [46]	CONSORT Statement. All of the items were judged as adequate (low risk of bias), inadequate (high risk of bias) or unclear (high risk of bias).	If the data of the variables were continuous, analyses were performed with calculations of the difference of the mean values and the standard deviation. Sub-Group analyses were performed according to the cues used (visual, auditory, sensory, combined and verbal instruction) and the outcomes (step length, cadence, speed, stride length and UPDRS III).	Gait analysis Auditory cues:increased step length (*p* = 0.03) and speed (*p* < 0.00001). Auditory cues during overground and treadmill gait compared with no cues: improvements in speed (*p* = 0.05) and cadence (*p* = 0.003), but no statistically significant difference for stride length (*p* = 0.89). Only one study reported significant reduction in freezing episodes after the intervention (*p* = 0.04)
		Heterogeneity was assessed in its clinical variations, and statistical methodology according to the principles of the Cochrane Collaboration. There was methodological heterogeneity in respect to the types of cues, but there was no compromise of the meta-analyses presented (≤40%). Statistical heterogeneity was assessed using the test I^2^ (“0% to 40%: might not be important; 30% to 60%: may represent moderate heterogeneity; 50% to 90%: may represent substantial heterogeneity; 75% to 100%: considerable heterogeneity”). The studies included were highly heterogeneous. Some variables could not be evaluated in all the comparisons because the outcomes were heterogeneous	
		All of the included studies were analyzed for risk of bias with a standardized table (Review Manager 5.2—Revman 5.2).	
Spaulding et al., 2013 [26]	Appraisal declared but the used tool was not specified	Quantitative information was collected from each study: means and SDs or other statistical measures of difference before and after cueing, cadence, stride length, and velocity.	Cadence:auditory cueing resulted in a significant increase in cadence (Hedge g: 0.556; 95% CI, 0.291–0.893). Stride length: auditory cueing resulted in a significant improvement (Hedge g: 0.554; 95% CI, 0.072–1.036). Gait velocity:auditory cueing resulted in a significant improvement (Hedge g: 0.544; 95% CI, 0.294–0.795).
		The data were synthesized using a meta-analysis, with the studies grouped based on cue type. The results from the meta-analysis were displayed in a random effects forest plot. An analysis based on random effects was used. It was chosen Hedge g is an effect size estimate.	
		The forest plots also provided statistics regarding the consistencies and strengths of the findings from each study and the overall effect when the results of the various studies were combined.	

RCT = randomized controlled trials; CRCT = cluster randomized controlled trials; CCT = controlled clinical trials; RAS = Rhythmical Auditory Stimulation; g = Hedge’s g significance; Hedges’ g; ES = Effect Size; CI = Confidence Interval.

**Table 4 brainsci-11-00685-t004:** Conclusions and limitations of selected studies.

Authors, Year	Conclusions	Limitations
Cassimatis et al., 2016 [44]	Adding sensory cues to a rehabilitation program resulted in higher levels of improvement in ADL performance measured as UPDRS II	The limited number of studies included in the meta-analysis
	Statistically significantly larger improvements on average for the UPDRS II were observed in the intervention group compared to the control group after treatment (*p* = 0.011)	
	Pre-post treatment means of UPDRS scores intervention 13.4–10.4 control 14.7–13.5	
Ghai et al., 2018 [45]	Evidence for positive effects of RAS on spatiotemporal gait parameters in PD patients.	The review attempted to be as comprehensive as possible. It mixes studies with different design, i.e., investigating both training effects and acute effects of RAS.
	Overall, 88% of studies reported beneficial effects of RAS on gait parameters with significant small-to-large standardized effects: stride length (g: 0.42) gait velocity (g: 0.23), cadence (g: −0.13).	Sometimes difficult to read, some errors in reporting references, numbers of included studies and values of ES do not always match.
Rocha et al., 2014 [46]	The present review corroborates the evidence that cues can promote significant gains in the PD gait, improve psychomotor performance and might reduce freezing episodes. Auditory cues could cause significant improvements in the step length and speed of the gait. Cadence improves with the use of auditory cues compared with interventions without the use of cues and compared to the use of cues (any type) during treadmill gait training versus gait training on the floor without cues.	Different treatment methodology regarding to stimulus intensity. Neither of them has the same number of therapies, distribution of therapies on the week, period of intervention and/or therapy time.
Spaulding et al., 2013 [26]	The findings of this meta-analysis infer that auditory forms of cueing are effective in improving important kinematic gait parameters, as cadence, velocity, and stride length.	The number of included studies and sample size were small. There were many different paradigms of cueing within each type, and therefore it is difficult to generalize the information to the rehabilitation setting. The study does not include use of gait assistive devices, which confines our results to people who had milder gait impairments.

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
