# Peer review of "The Impact of Exercise Intervention with Rhythmic Auditory Stimulation to Improve Gait and Mobility in Parkinson Disease: An Umbrella Review"

_brainsci, 2021, doi:10.3390/brainsci11060685_

Round 1

Reviewer 1 Report

The current paper reviewed two systematic reviews and meta-analyses, one systematic review and one meta-analysis on the effectiveness of rhythmic-cued exercise on gait, mobility and Activities of Daily Living (ADL) in people with Parkinson's disease. It concluded that this type of exercise should be incorporated into PD rehabilitation but specific guidelines for its application is to be established. My general comments are that the current paper lack full appraisal of the selected review papers, does not appear to add much to the exiting literature, and because information is not well organized (in Discussion, Conclusions, and Limitations), it is hard to follow. It is not clear whether reviewing the selected review papers would achieve what the authors aimed to do (i.e., to help to establish the exercise guidelines), given only one review paper presents the relevant information. The authors reported that a slight level of overlapping references in the selected review papers. This indicates that these review papers have different aims to present the published studies.  

Abstract:

  • Please spell out ADL.
  • Please correct "one a systematic review and one a meta-analysis"

Introduction

p.g. 2  "It remains unclear... but the short-term benefits of exercise training programs suggest this possibility.", which needs references to support this claim.

p.g. 2  Why do the authors focus on RAS is unclear as other types of physical exercise seem beneficial, too. Is the rhythmic-cued exercise better than the others? 

Materials and Methods

2.2 Search strategy and study selection

  • Please spell out CCTS and QRCT.

Table 2 - Rocha et al., 2014

  • Please correct "7, 6 RCTs"

Discussion

  • The first paragraph can be combined with other limitations of the reviewed papers discussed in Limitations.
  • The readers would expect full appraisal of the selected review papers presented by the authors in this section. However, this section mostly discusses the literature outside of the reviewed papers.
  • In Paragraph 5, some explanations why the authors think Rocha et al did not find the significant effect on cadence would be helpful. 
  • In Paragraph 8, it is unclear what the authors were trying to convey from the part of “The use of auditory cues to …scaling movement time.”
  1. Conclusions
  • In the first paragraph, the overall conclusion on the RAS intervention made by the authors is nothing new.
  • In the first paragraph, the authors say that “The use of an auditory external stimulus …may facilitate the attention on the control of movements.” There is no previous discussion about attention and motor control.
  • In the second paragraph, please add references in the paragraph “In fact, ….the individual’s attention capacity.”
  • In the second paragraph, it would be helpful to expand “multi-component approach to exercise” to present the authors’ perspective for future studies.

5.1 Limitations

  • It is confusing this section discusses the limitations of both current paper and reviewed paper. The limitations of the reviewed paper can be discussed in Discussion (i.e., “these often presented inadequate description…. regarding the applied exercise”.
  • The authors can expand the paragraph “The first level studies summarized …” to discuss more details about the selected review papers. This also can be included in Discussion.

Author Response

We wish to thank both reviewers for their useful comments which we have done our best to follow. We have hopefully improved the organization of the paper by either moving paragraphs as suggested, or by removing excessive information. We hope it will be now “lighter” and easier to follow. We hope the reviewer will agree that the work has improved.

All the changes in the manuscript have been highlighted: in red the part of the manuscript which have been moved from one section to another, in green those completely new, in blue strikethrough those removed.

REPLY to REVIEWER 1

The current paper reviewed two systematic reviews and meta-analyses, one systematic review and one meta-analysis on the effectiveness of rhythmic-cued exercise on gait, mobility and Activities of Daily Living (ADL) in people with Parkinson's disease. It concluded that this type of exercise should be incorporated into PD rehabilitation but specific guidelines for its application is to be established.

My general comments are that the current paper lack full appraisal of the selected review papers,

Please see reply to the same comment relative to the discussion section

…does not appear to add much to the exiting literature, and because information is not well organized (in Discussion, Conclusions, and Limitations), it is hard to follow. It is not clear whether reviewing the selected review papers would achieve what the authors aimed to do (i.e., to help to establish the exercise guidelines), given only one review paper presents the relevant information.

Our aim was to provide a base for future development of guidelines prescription (“to guide future development of guidelines for Rhythmic auditory stimulation” (aims line page 4 line 5)). However, this was not completely possible. Though RAS for gait and mobility is described as a particularly beneficial tool for PD patients in a significant number of reviews, the available information is not yet systematically organised to delineate clear guidelines. Possibly the last sentence of the abstract may have created some misunderstanding, therefore we have changed it to be more in line with the aims in the text.

Anyhow, we agree with the reviewer, we could not add much to the existing literature because, despite the apparent large number of reviews and meta-analyses on the use of RAS, we found that most of them did also include studies mixing different types of exercise (dance or physiotherapy), or different exercise effects (acute or long-term) or different perceptive stimuli (visual with auditory) and could not be included. Therefore, strictly regarding the use of RAS on gait and mobility, we found very little material. According to us, there is still a lack of information on to how to modulate the “overload” of RAS because primary level research is too varied in terms of frequency, intensity, duration, patients’ characteristics (on-off medication/ level of disease). An additional challenge is posed by the identification and definition of outcomes. While kinematic measures of gait are standardised, difficulties arise when meta-analyses or systematic reviews have similar but not identical definition of outcomes such as mobility, and some may have looked at performance of functional tasks while others at questionnaires or scales such as UPDRS.

Despite this we believe that our work is still important as it suggests the need to perform more primary level research with a clear structure to address the uncertainties on the methods for RAS application to physical exercise prescription. Clarifying these aspects would ideally lead to establish guidelines for practice.

The authors reported that a slight level of overlapping references in the selected review papers. This indicates that these review papers have different aims to present the published studies.

The authors are not completely clear about this point. In any case, we have added a comment in respect to the overlap and modified the related sentence. The calculation of overlap was performed as described by Pieper et al., (2014) because of similarities of the selected reviews in the topic of investigation and interventions to avoid the inclusion of primary studies more than once which may result in biased findings (page 6).

As reported in the tables the aims, outcomes and interventions were similar:

Cassimatis To verify the effects of external sensory cues in improving functional performance of ADL, walking, and daily tasks in PD

Ghai To analyze the effects of rhythmic auditory cueing on gait and postural performance in patients affected by Parkinson’s disease.

Rocha Evaluate the benefits of external cues on the gait of PD patients and their impact on the quality of life, freezing and psychomotor performance (which translated into the effects on step and stride length speed of gait, cadence and UPDRS, with one study analyzed freezing).

Spaulding To compare the relative efficacy of visual versus auditory cueing on gait among individuals with Parkinson’s disease (PD).

Abstract:

  • Please spell out ADL.

ADL has been written in full

  • Please correct "one a systematic review and one a meta-analysis"

Corrected as requested

Introduction

p.g. 2  "It remains unclear... but the short-term benefits of exercise training programs suggest this possibility.", which needs references to support this claim.

Reference added as requested (page 2). Rafferty MR, Schmidt PN, Luo ST, Li K, Marras C, Davis TL, Guttman M, Cubillos F, Simuni T. Regular Exercise, Quality of Life, and Mobility in Parkinson's Disease: A Longitudinal Analysis of National Parkinson Foundation Quality Improvement Initiative Data. J Parkinsons Dis 2017;7(1):193-202. doi: 10.3233/JPD-160912.

p.g. 2  Why do the authors focus on RAS is unclear as other types of physical exercise seem beneficial, too. Is the rhythmic-cued exercise better than the others? 

As described in different reviews (as cited in the manuscript), for gait and mobility exercise with RAS seems particularly beneficial with respect to exercise without RAS. The mechanisms behind the potentially greater benefits have been described briefly in the introduction and then in the discussion. We have added a short sentence in support of the importance of RAS before the aims (end of page 3). We have re-organized the introduction also following the 2nd reviewer comments and moved part of the discussion in the introduction. This should clarify the importance of RAS for PD patients.

Materials and Methods

2.2 Search strategy and study selection

  • Please spell out CCTS and QRCT.

Corrected as requested (page 4)

Table 2 - Rocha et al., 2014

  • Please correct "7, 6 RCTs"

Corrected as requested “7 studies: 6 RCTs, 1 QRCT”

Discussion

  • The first paragraph can be combined with other limitations of the reviewed papers discussed in Limitations.

This first paragraph was part of the authors’ guidelines and not erroneously canceled. We are sorry of the mistake. It has been removed

The readers would expect full appraisal of the selected review papers presented by the authors in this section. However, this section mostly discusses the literature outside of the reviewed papers.

We thank the reviewer for the useful comment. We have removed some of the literature not strictly concerned to the reviewed papers and some of it has been moved in the introduction. Full quality appraisal of the reviewed paper is reported in the results section, and at the beginning of the discussion we have added critical comments regarding the results of the appraised reviews. We hope this will satisfy the query (page 19 in green).

  • In Paragraph 5, some explanations why the authors think Rocha et al did not find the significant effect on cadence would be helpful. 

Comments have been added in the discussion, please see above comment

  • In Paragraph 8, it is unclear what the authors were trying to convey from the part of “The use of auditory cues to …scaling movement time.”

Scaling has been substituted with grading, possibly it better renders the idea that RAS seems to help in limiting “jerky” movements (page 3)

  1. Conclusions
  • In the first paragraph, the overall conclusion on the RAS intervention made by the authors is nothing new.

The sentence has been removed

  • In the first paragraph, the authors say that “The use of an auditory external stimulus …may facilitate the attention on the control of movements.” There is no previous discussion about attention and motor control.

The reviewer is right, it is not a topic covered in the review, the sentence has been removed, the whole discussion has been shortened and restricted to the topic of the review.

  • In the second paragraph, please add references in the paragraph “In fact, ….the individual’s attention capacity.”

Reference added as requested (page 21 line 2)

Kelly VE, Eusterbrock AJ, Shumway-Cook A. A review of dual-task walking deficits in people with Parkinson's disease: motor and cognitive contributions, mechanisms, and clinical implications Parkinsons Dis . 2012;2012:918719.

  • In the second paragraph, it would be helpful to expand “multi-component approach to exercise” to present the authors’ perspective for future studies.

A comment on the authors’ idea of multi-component approach to exercise has been added (page 21 in green)

 5.1 Limitations

  • It is confusing this section discusses the limitations of both current paper and reviewed paper. The limitations of the reviewed paper can be discussed in Discussion (i.e., “these often presented inadequate description…. regarding the applied exercise”.

This section describing the limitation of the included studies has been moved to the discussion as requested

  • The authors can expand the paragraph “The first level studies summarized …” to discuss more details about the selected review papers. This also can be included in Discussion.

All the points regarding the reviewed papers originally in the limitations section have been moved in the discussion section, which we feel now more critically discuss the lack of findings (page 20).

Reviewer 2 Report

Thank you for the opportunity to review this interesting study that aims at identifying specific evidence to guide future development of guidelines for RAS intervention. This is still lacking despite the fact that several systematic reviews and meta-analyses have been conducted the last decade, and the study is relevant. The authors conclude that this is indeed a difficult area, and that more research is needed. I think that this research is sound and is based on logic which makes it very trustworthy.

Although the study is in general very well-written, I have a few minor suggestions to improve the contents before publishing.

  1. The text is quite extensive, and I feel it would improve the readability to reduce some of it. For example, in the background there are many facts about the benefits of physical exercise and physical training and consequences from sedentary behavior that has little to do with your specific aim. It would be better to replace this with some facts about the underlying mechanisms of RAS from the discussion section (which does not really belong there). The discussion section should, in my opinion, mainly focus on your findings, i.e., the results from the review. The long section about the underlying mechanisms could also preferably be somewhat shortened, although it is indeed highly interesting and well written.

  2. I think the second paragraph on page 4 (starting with “With respect to the methodological quality”) should be moved to the end of the results section instead, since this is actually the results of your review. This would be more logical, since the method section should only describe what you are planning to do. For me, it is strange to start by telling us (i.e., the readers) about the methodological quality before we are informed about how many studies you have included in your review.

  3. Please be consequent with spelling of the data bases (correct should be: PubMed and Embase). You use different spellings in the methods section, the results section and the flow chart.

  4. I understand that this is a work manuscript, but I hope that the Table 2 will be printed horizontally instead – this would improve the readability a lot.

  5. The discussion section (4) starts with a paragraph that seems to be some sort of instructions?

  6. Please be consequent with writing the concept rhythmical auditory stimulation and/or the acronym RAS throughout the manuscripts – you are using both variants when you should be using RAS continuously.

  7. Is it necessary to mention that the use of auditory cues have been used since the 60s? I think not.

  8. The conclusion is far too extensive and contains facts that have not previously been mentioned within the results section. I think the second paragraph should be removed to the discussion section altogether. I also think that you should end the conclusion with the final two sentences on page 21 – this will be a perfect ending to a conclusion.

  9. In my opinion, the limitations section (5.1) should only refer to the present study’s limitations, not the limitations of the included reviews (which belongs to the methodological considerations/quality review). This may, for example, be about the choices you have made concerning databases and/or search strategies. Most of your present text is about the limitations of the included studies, and this should be discussed in the results section.

Author Response

REPLY to REVIEWER 2

Thank you for the opportunity to review this interesting study that aims at identifying specific evidence to guide future development of guidelines for RAS intervention. This is still lacking despite the fact that several systematic reviews and meta-analyses have been conducted the last decade, and the study is relevant. The authors conclude that this is indeed a difficult area, and that more research is needed. I think that this research is sound and is based on logic which makes it very trustworthy.

Although the study is in general very well-written, I have a few minor suggestions to improve the contents before publishing.

We would like to thank the reviewer for the positive comments. We have done our best to follow all the suggestions and we hope the paper has improved and lightened

  1. The text is quite extensive, and I feel it would improve the readability to reduce some of it. For example, in the background there are many facts about the benefits of physical exercise and physical training and consequences from sedentary behavior that has little to do with your specific aim. It would be better to replace this with some facts about the underlying mechanisms of RAS from the discussion section (which does not really belong there). The discussion section should, in my opinion, mainly focus on your findings, i.e., the results from the review. The long section about the underlying mechanisms could also preferably be somewhat shortened, although it is indeed highly interesting and well written.

We have removed from the background the information regarding the benefits of exercise and the consequences of sedentary behaviour. We have also moved and shortened parts of the discussion in the introduction in order to clarify from the beginning the underlying mechanisms and the potential advantages of RAS with respect to exercise without RAS (page 2 and 3)

  1. I think the second paragraph on page 4 (starting with “With respect to the methodological quality”) should be moved to the end of the results section instead, since this is actually the results of your review. This would be more logical, since the method section should only describe what you are planning to do. For me, it is strange to start by telling us (i.e., the readers) about the methodological quality before we are informed about how many studies you have included in your review.

Paragraph has been moved in the results (page 7)

  1. Please be consequent with spelling of the data bases (correct should be: PubMed and Embase). You use different spellings in the methods section, the results section and the flow chart.

Corrected as requested

  1. I understand that this is a work manuscript, but I hope that the Table 2 will be printed horizontally instead – this would improve the readability a lot.

  2. The discussion section (4) starts with a paragraph that seems to be some sort of instructions?

This first paragraph was not erroneously canceled, it was part of the authors’ guidelines. We are sorry of the mistake. It has been removed.

  1. Please be consequent with writing the concept rhythmical auditory stimulation and/or the acronym RAS throughout the manuscripts – you are using both variants when you should be using RAS continuously.
    Corrected throughout the manuscript
  2. Is it necessary to mention that the use of auditory cues have been used since the 60s? I think not.

Sentence has been deleted

  1. The conclusion is far too extensive and contains facts that have not previously been mentioned within the results section. I think the second paragraph should be removed to the discussion section altogether. I also think that you should end the conclusion with the final two sentences on page 21 – this will be a perfect ending to a conclusion.

We agree that the conclusion was too long, we have shortened it and moved the second paragraph in the discussion.

  1. In my opinion, the limitations section (5.1) should only refer to the present study’s limitations, not the limitations of the included reviews (which belongs to the methodological considerations/quality review). This may, for example, be about the choices you have made concerning databases and/or search strategies. Most of your present text is about the limitations of the included studies, and this should be discussed in the results section.

Yes, we agree, we have moved the limitations of the previous work in the discussion where we have, as suggested by the 1st reviewer, made more comments on the results of the presented reviews (page 20)

Round 2

Reviewer 1 Report

The manuscript has been significantly improved and reads well. The rationale for the current review is clearly mentioned in Introduction.

I have a few minor points that can be revised. 

  1. pg 3, 3rd paragraph: is "for" [22] typo?
  2. pg3, 3rd paragraph: please put a citation (citations) after "other gait and balance related difficulties." 
  3. pg3, 4th paragraph: please put proper citations for "This process depends on...supplementary motor area". Has entrainment been demonstrated in these areas ?
  4. pg6, 3.2, 1st para, 2nd sentence: I would suggest that "for reviews of reviews" may be reworded. 

Author Response

We appreciate the further work and comments of the reviewer who we wish to thank. We have implemented the requested changes.

The manuscript has been significantly improved and reads well. The rationale for the current review is clearly mentioned in Introduction.

I have a few minor points that can be revised. 

  1. pg 3, 3rd paragraph: is "for" [22] typo?

Yes, it was a typo, it has been removed

  1. pg3, 3rd paragraph: please put a citation (citations) after "other gait and balance related difficulties."

Citation has been added

Kadivar Z, Corcos DM, Foto J, Hondzinski JM. Effect of step training and rhythmic auditory stimulation on functional performance in Parkinson patients Neurorehabil Neural Repair 2011 Sep;25(7):626-35.

  1. pg3, 4th paragraph: please put proper citations for "This process depends on...supplementary motor area". Has entrainment been demonstrated in these areas ?

Sentence has been slightly changed and properly citated (Koshimori and Thaut, 2018)

  1. pg6, 3.2, 1st para, 2nd sentence: I would suggest that "for reviews of reviews" may be reworded. 

We have changed it to “umbrella reviews”